# Synthesis and In Vitro Photocytotoxicity of 9-/13-Lipophilic Substituted Berberine Derivatives as Potential Anticancer Agents

**DOI:** 10.3390/molecules25030677

**Published:** 2020-02-05

**Authors:** Hong-Jhih Lin, Jinn-Hsuan Ho, Li-Chen Tsai, Fang-Yu Yang, Ling-Ling Yang, Cheng-Deng Kuo, Lih-Geeng Chen, Yi-Wen Liu, Jin-Yi Wu

**Affiliations:** 1Department of Microbiology, Immunology and Biopharmaceuticals, College of Life Sciences, National Chiayi University, Chiayi 60004, Taiwan; s1030730@mail.ncyu.edu.tw (H.-J.L.); s1003500@mail.ncyu.edu.tw (L.-C.T.); s1060645@mail.ncyu.edu.tw (F.-Y.Y.); lgchen@mail.ncyu.edu.tw (L.-G.C.); ywlss@mail.ncyu.edu.tw (Y.-W.L.); 2Department of Chemical Engineering, College of Engineering, National Taiwan University of Science and Technology, Taipei 10607, Taiwan; jhho@mail.ntst.edu.tw; 3Department of Pharmacognosy, School of Pharmacy, College of Pharmacy, Taipei Medical University, Taipei 11031, Taiwan; llyang@tmu.edu.tw; 4Division of Chest Medicine, Department of Internal Medicine, Changhua Christian Hospital, Changhua 50006, Taiwan; cdkuo23@gmail.com; 5Department of Medical Research, Taipei Veterans General Hospital, Taipei 11217, Taiwan

**Keywords:** berberine, lipophilic substituent, anti-cancer activity, photocytotoxicity, reactive oxygen species

## Abstract

The objective of this study was to synthesize the 9-/13-position substituted berberine derivatives and evaluate their cytotoxic and photocytotoxic effects against three human cancer cell lines. Among all the synthesized compounds, 9-*O*-dodecyl- (**5e**), 13-dodecyl- (**6e**), and 13-*O*-dodecyl-berberine (**7e**) exhibited stronger growth inhibition against three human cancer cell lines, (HepG2, HT-29 and BFTC905), in comparison with structurally related berberine (**1**). These three compounds also showed the photocytotoxicity in human cancer cells in a concentration-dependent and light dose-dependent manner. Through flow cytometry analysis, we found out a lipophilic group at the 9-/13-position of berberine may have facilitated its penetration into test cells and hence enhanced its photocytotoxicity on the human liver cancer cell HepG2. Further, in cell cycle analysis, **5e**, **6e,** and **7e** induced HepG2 cells to arrest at the S phase and caused apoptosis upon irradiation. In addition, photodynamic treatment of berberine derivatives **5e**, **6e,** and **7e** again showed a significant photocytotoxic effects on HepG2 cells, induced remarkable cell apoptosis, greatly increased intracellular ROS level, and the loss of mitochondrial membrane potential. These results over and again confirmed that berberine derivatives **5e**, **6e,** and **7e** greatly enhanced photocytotoxicity. Taken together, the test data led us to conclude that berberine derivatives with a dodecyl group at the 9-/13-position could be great candidates for the anti-liver cancer medicines developments.

## 1. Introduction

Although significant advances have been made in preventing and treating cancer, the mortality rate remains to be unacceptably high. Meanwhile, many of the current chemotherapeutics are limited by significant side effects, unpredictable efficacies, and/or acquired resistances [1,2,3]. 

Human hepatocellular carcinoma (HCC) and colorectal carcinoma are the most common types of malignancy in sub-Saharan Africa and Southeast Asia, like Taiwan and China. It is also the fifth most common and lethal cancer worldwide. Most of all, HCC is the most frequent malignant tumor and the second leading cause of cancer-related death in Taiwan [4]. The hepatoma cells have high proliferative and metastasis rates. It is usually treated by surgical removal, followed by chemotherapy [5,6]. Doxorubicin is widely used in HCC chemotherapy. However, the agent failed to significantly improve the 5-year survival rate of the patients partly for their severe side effects, which, among others, include hepatic dysfunction and heart failure. As a group, we have been pursuing projects aimed at developing novel cancer therapeutics that improve the prognosis of the disease with less adverse side effects.

Many natural products have been rich resources for potential drug development. Alkaloids, among which are a well-known nitrogen-containing group of diverse plant secondary metabolites [7]. Especially, among them, berberine (**1**, Figure 1), is an isoquinoline alkaloid isolated from many medicinal herbs, such as *Coptis chinensis, Berberis aristata*, and *Phellodendron amurense*. It is widely used as traditional Chinese medicine [8,9,10]. Ayurvedic medicine in China and India [11]. It has a wide range of biochemical and pharmacological potencies including antibacterial, anti-inflammatory, and anti-cancer effects [12,13]. Further, berberine has a long history of medical application and was widely used as an anti-cancer drug to treat hepatoma [14], breast cancer [15], bladder cancer [16], and colon cancer [17]. Furthermore, berberine has significant cytotoxicity against human cancer cell lines by triggering a significant increase in G1 arrest in the cell cycle phase of HepG2 cells [18]. It also induced cell cycle arrest at the G2/M phase and caused apoptosis of human gastric cancer SNU-5 cells [19]. However, owing to its hydrophilic nature, or lower bioavailability, berberine is absorbed poorly in intestines. This results in a low efficacy in suppressing cancer cell growth [20]. Besides, the anti-cancer activity of berberine and its derivatives had recently been shown as an inhibitor of topoisomerase I and II [21,22].

In addition, a recent study reported various analogs of berberine on the 9-/13-position using different chain lengths and terminal amino groups were synthesized and evaluated on the DNA-binding affinity or as G-quadruples stabilizing ligands [23,24,25,26]. The study also showed the introduction of a lipophilic substituent into the 9-*O*-position of the berberine backbone and enhanced the anti-proliferative activities against human cancer cell lines [27,28,29,30]. However, structural modification on the 13-position of the berberine backbone has rarely been reported [31]. To explore its intestinal absorption and inhibitory function, in this study, we modified berberine analogs and evaluated its photo-induced cytotoxicity in vitro on human liver HepG2, bladder BFTC905 and colon HT29 cancer cell lines. The type of functional groups, *n*-alkyl derivatives of berberine at the 9- or 13-position, was investigated for the structure-activity relationship (SAR).

## 2. Results and Discussion

### 2.1. Chemistry

In general, the desired berberine derivatives were synthesized with long-chain *n*-alkyl bromide through 9-/13-alkylation. As shown in Scheme 1, in specific, synthesis of **5a**–**e**, **6a**–**e,** and **7a**–**e** was carried out by reacting berberrubine (**2**) with selected *n*-alkyl bromide in dry acetone or acetonitrile using anhydrous potassium carbonate (K_2_CO_3_) as a base under nitrogen for 8‒24 h according to the reported procedure [27]. The reduction of berberine (**1**) using sodium borohydride (NaBH_4_) gave intermediate **3** in 74% yield. The addition of long-chain *n*-alkyl group to berberrubine (**2**) and 13-hydroxyberberine (**4**) gave the products of 9-/13-*O*-position derivatives in 40–86% yield. The compound **3** was reacted with *n*-alkyl aldehyde in an ethanol and acetic acid solvent mixture, and then acidified with 2 N HCl to yield the desired 13-position derivatives **6a**–**e** in 29%–76% yield [31].

Further, the reaction mixture was refluxed for 8–24 h, and the progress of the reaction was monitored by TLC. The mixture was then cooled to room temperature, and the solvent was removed under reduced pressure. The crude product was chromatographed on silica gel column and eluted with CH_2_Cl_2_/MeOH (15/1) solvent to afford the desired 9-/13-position berberine derivatives **5a**–**e**, **6a**–**e,** and **7a**–**e** in quantitative yields (Scheme 1). All compounds were purified by flash column chromatography and were thoroughly characterized by ^1^H, ^13^C, 2D NMR spectroscopy, liquid chromatography-mass spectroscopy, and high-resolution mass spectroscopy. The full signal assignment of ^1^H and ^13^C NMR techniques including DEPT, COSY, HSQC, and HMBC analyses. All of the final compounds showed >98% purity by HPLC analysis.

### 2.2. Biological Activity

#### 2.2.1. In Vitro Anti-Proliferative Activity by MTT Assay

The cytotoxic effect of the compounds was tested by the 3-(4,5-dimethylthiazol- 2-yl)-2,5-diphenyltetrazolium bromide (MTT) assay [32] and rated by half-maximal inhibitory concentration (IC_50_). IC_50_ is a measure of the effectiveness of a compound in inhibiting biological or biochemical function. The cytotoxic effects of the 15 9-*O*-, 13- or 13-*O*-position substituted berberine derivatives were tested against three human cancer cell lines and one normal murine embryonic liver cell line. The cancer cells under study were HepG2 (liver carcinoma), HT29 (colon carcinoma), and BFTC905 (bladder carcinoma). As shown in Table 1, we found firstly, as a trend, the anti-cancer activity of these compounds increased with the alkyl chain length, or lipophilic characteristics, of the substitutes. Thus, the remarkable anti-cancer activity of the berberine derivatives with 9-/13-position substitution of the dodecyl moiety can be attributed to their enhanced bioavailability and cell membrane permeability through the increased lipophilicity [33].

As was shown in Figure 2, the cytotoxicity effect of three bioactive derivatives; compounds **5e** (9-*O*-dodecylberberine), **6e** (13-dodecylberberine), and **7e** (13-*O*-decylberberine). In essence, all showed significant cell growth inhibition with the IC_50_ values of 0.32 ± 0.08, 0.77 ± 0.18 and 0.83 ± 0.30 μM, respectively, against human liver carcinoma HepG2 cells. Compared to berberine (**1**)’s IC_50_ value of >40 μM, these three derivatives evinced 48- to 125-fold stronger cytotoxicity potency. In other words, through a simple structure-activity relationship (SAR) analysis, we have shown that the cell growth inhibitory potency closely related to the chain length of *n*-alkyl groups. Meanwhile, to check potential side effects, we treated normal Madin–Darby Canine Kidney (MDCK) cell lines with berberine (**1**) and compounds **5e**, yet detected no significant cell death (data not shown). Furthermore, only a marked effect on cell death (under 20%) was observed at the maximum concentration (4 μM) of **5e** and **6e** after 48 h treatment, except for berberine (**1**), which exhibited slight toxic. It appeared that compounds **5e**, **6e** and **7e** are cytotoxic to human colon cancer cells with no significant adverse effects on normal MDCK cells.

To confirm the findings above, we calculated partition coefficient (clog*P*) values of 9-/13-substituted berberine derivatives and compared it with the cytotoxic activity. As shown in Table 1, clog*P* values correlated well with tested anti-cancer activity for 9-/13-substituted berberine derivatives. These results just once again verified that the lipophilicity or the chain length of the *n*-alkyl moieties of 9-/13-substituted berberine derivatives played a very important role in their anti-cancer activity.

Further, the positive clog*P* values of berberine derivatives **5e**, **6e**, and **7e** showed that the introduction of the long-chain *n*-alkyl groups to the berberine scaffold at the 9-*O*-, 13-, and 13-*O*-position will moderately augment the lipophilicity of berberine, while *n*-alkyl substituents at the 9-*O*-, 13-, and 13-*O*-positions of berberine remarkably increased the lipophilicity of berberine (Table 1). It is also worth noting, berberine derivatives with a moderate carbon chain length (**5e**, **6e,** and **7e**) would significantly enhance its lipophilicity and yield much more favorable transmembrane permeability. In general, the clog*P* value is a good index in estimating the distribution of drugs within the cells. Thus, hydrophobic drugs with high 1-octanol/water partition coefficients are preferentially distributed to hydrophobic compartments such as the lipid bilayers of cells while hydrophilic drugs (low 1-octanol/water partition coefficients) preferentially are found in aqueous compartments such as blood serum.

#### 2.2.2. Absorption and Emission Spectra of Berberine (**1**) and its Derivatives (**5e**, **6e,** and **7e**)

The absorption and fluorescence spectra of berberine (**1**) and its derivatives (**5e**, **6e,** and **7e**) in MeOH were measured (Figure 3). The absorption spectrum of the derivative **6e**, with the dodecyl group at the 13-position of berberine, exhibited a moderate bathochromic shift (ca. 8.3 nm) than the other compounds. Its absorptivity was obviously significantly decreased compared with that of berberine (**1**) and other derivatives. Figure 3B showed the emission spectra of berberine (**1**) and its derivatives (**5e**, **6e,** and **7e**) in MeOH (10 μM). All gave the excitation wavelength at 420 nm with the emission maxima of 529, 531, 516, and 524 nm, respectively. The data suggested that compounds **6e** and **7e** have higher energy of the emission spectra in MeOH, which, in turn, may enhance its photocytotoxicity in photodynamic therapy.

#### 2.2.3. Intracellular Uptake of Berberine (**1**) and Its Derivatives (**5e**, **6e,** and **7e**)

Intracellular uptake of an anti-cancer agent is to check if the medicine would be able to target key organelles of the cell. Using time-dependent HPLC analysis, we studied the cellular uptake of berberine (**1**) and its derivatives. First, cells were treated with 10 µM of berberine (**1**) and its derivatives at different time points, viz. 0.25, 1, 2, 4, and 8 h. The emission of the berberine (**1**) and its derivatives inside the cells was monitored (Figure 4). The percentage incorporation of berberine (**1**) and its derivatives (**5e**, **6e,** and **7e**) into the cells is shown in Figure 4. The data suggested that both the berberine (**1**) and its derivatives internalized into the cancer cells to the same extent with similar rates of internalization. A nearly 100% uptake was observed when the cells were incubated with the compounds for 2-4 h. As shown in Figure 4, we found that the intracellular concentrations of berberine (**1**) and its three derivatives increased rapidly in 1 h, reached a plateau at 2 h, then stayed at the plateau for another 6 h. The uptake amounts of compounds **5e**, **6e**, and **7e** were significantly higher than those of berberine (**1**). This observation confirmed our calculated higher partition coefficients (clog*P*) of compounds **5e**, **6e,** and **7e**. Thus, berberine derivatives (**5e**, **6e**, and **7e**) with moderate carbon chain length showed the most significant enhancement in lipophilicity, implying more preferable *trans*-membrane permeability than berberine.

#### 2.2.4. Photocytotoxicity of Berberine (**1**) and Its Derivatives (**5e**, **6e,** and **7e**)

To determine the photo-induced cytotoxicity of berberine (**1**) and its derivatives (**5e**, **6e**, and **7e**) in HepG2 cells, proliferation assay was performed. Exponentially growing HepG2 cells were exposed for 10 min to visible light (420 nm, 5.9 mW/cm^2^) in the presence of increasing concentrations of berberine (**1**) and its derivatives. Subsequently, the effect of the treatments on cell proliferation was determined by the cell counts right after damage and after 20 h of incubation in fresh media, free of berberine derivatives. It is notable that, as shown in Figure 5, the berberine derivatives caused a quite similar dose-dependent reduction of cell proliferation under visible light irradiation. On the contrary, no growth inhibition was seen by the irradiation alone (see data point at 0 μM of berberine derivatives) and in non-irradiated cells pre-incubated with 0.5 μM of berberine derivatives (see open symbols in Figure 5).

In light of the dark cytotoxicity findings described above, we extended our study further on the anti-proliferation efficacy of berberine (**1**), compounds **5e**, **6e,** and **7e** against HepG2 cells over a 24 h period with data on berberine (**1**) as the base case. As shown in Table 2, first of all, the dark cytotoxicity of compounds **5e**, **6e,** and **7e** demonstrated, once again, an impressive 125-, 52-, and 48-fold, respectively, stronger bioactivity than that of berberine (**1**) in vitro. In addition, the growth inhibition of HepG2 cells induced by these three samples was a concentration- and light dose-dependent manner.

The cytotoxicity of berberine (**1**) and its derivatives (**5e**, **6e,** and **7e**) evaluated after incubation with HepG2 cells in the dark as well as after exposure to visible light (420 nm) for 10 min. Further, the photo-induced cytotoxic effects of berberine (**1**) and its derivatives on three human cancer cell lines was assessed using a light dose of 3.6 J/cm^2^ at 420 nm wavelength, respectively. The resulting cells were irradiated with visible light (420 nm) for 10 min irradiation. After photo-irradiation, the cell viability was measured by MTT assay, of which these results are shown in Figure 5 along with the IC_50_ values in Table 2. The control experiment without light irradiation showed slight cytotoxicity (85% ~ 95% of the cell viability). The dark cytotoxicity of these berberine derivatives was evaluated in HepG2 cells at the concentration of 0.5 μM without photo-irradiation. We found that the berberine derivatives **6e** and **7e** showed no significant cytotoxicity in the dark (Figure 5), except over 1 μM. Furthermore, Figure 5 showed the cell survival ratio as a function of the concentrations of berberine derivatives. The IC_50_ value of **6e** was 0.15 μM, which was almost the same as that of **7e** (0.14 μM) and lower than that of **5e** (0.28 μM).

#### 2.2.5. Cell Morphological Assessment

The apoptotic cells change in morphology, such as chromatin condensation, nuclear shrinking, and DNA fragmentation. It can be characterized through Hoechst 33,258 stained cell nucleus [34] under a fluorescent microscope. Thus, to further study the role of apoptosis played in the superior cytotoxicity of compounds **5e**, **6e,** and **7e**, we incubated HepG2 cells in 0.5 μM of berberine (**1**), or its three derivatives separately. As shown in Figure 6A, while the nuclei of the cells were round in shape homogenously in the control without the testing compound, those treated with berberine (**1**), compound **5e**, **6e,** or **7e** showed typical morphological features of apoptosis such as cell blebbing and nuclear shrinkage. Figure 6B shows that HepG2 cells treated with 0.5 μM of compound **7e** stained with Hoechst 33,258 in dark or light, and examined under fluorescence microscopy for topical morphological changes, such as chromatin condensation and DNA fragmentation [34]. Evidently, the proliferation of the cancer cells was inhibited by the testing compounds in vitro. These results showed that compounds **5e**, **6e,** or **7e** induce apoptosis in HepG2 cells through irradiation.

#### 2.2.6. Cell Cycle Distribution Analysis Using Flow Cytometry

To probe the apoptotic effects of berberine (**1**) and its derivatives on cell cycle progression of HepG2 cells were treated once again with these compounds, separately, at 0.5 μM concentrations for 2 h after irradiation (420 nm, 10 min) for 24 h. Then, the samples were further treated with propidium iodide (PI) staining [35], and of which the sub-G1 phase was analyzed by flow cytometry. Besides, the cells with no test compound were the control of the experiment. Berberine (**1**) and its derivatives (**5e**, **6e,** and **7e**) showed significant photocytotoxic property in visible light at 420 nm wavelength. To find out whether the said property plays a role in the apoptosis of the cells, flow cytometric analysis of the HepG2 cells treated with berberine (**1**) and its derivatives (**5e**, **6e,** and **7e**) was carried out at 0.5 μM concentrations. Cells initially treated with the compounds in the dark and tested before and after being exposed to visible light (420 nm) and then stained with propidium iodide (PI) to observe any fragmented DNA or an increase in the sub-G1 population. As illustrated in Figure 7, the percentage of S phase increased by 32.71% and 37.58% in the dark and light for the control, respectively, in the sample with **5e** increased to 41.14%, 49.71% for **6e,** and 44.36% for **7e** compared to the 37.58% for the control in the light. The increased S phase percentage and the sub-G1 population in compounds **5e**, **6e,** and **7e** treatment suggested that **5e**, **6e,** and **7e** induced S phase arrest and caused apoptosis in HepG2 liver cancer cells. At first, berberine (**1**) and its derivatives (**5e**, **6e** and **7e**) being non-cytotoxic in the dark at 0.5 µM concentration, but it induced significant apoptosis in the cells at a concentration of 0.5 µM upon visible light irradiation for 24 h. In addition, the percentage of apoptotic cell populations increased with increasing compounds concentration (Figure 7). Lastly, a significant induction of apoptosis in the cells occurred even in the dark with an increasing concentration of derivatives.

#### 2.2.7. Measurement of Intracellular ROS Production by Irradiation

Several alkaloids induce apoptosis by generating singlet oxygen (^1^O_2_) in mitochondria were reported [36]. Using dichlorofluorescein diacetate (DCFH-DA) assay, we checked if berberine (**1**), compounds **5e**, **6e,** or **7e** could stimulate the generation of reactive oxygen species (ROS) after irradiation at visible light (420 nm) in HepG2 cells. We found that the fluorescence intensity of DCF in the cells was right-shifted after the cells were treated with all four compounds in a 0.5 μM concentration upon irradiation. Shown in Figure 8A, berberine (**1**), compounds **5e**, **6e**, or **7e** all stimulated to release intracellular ^1^O_2_ from HepG2 cells and exhibited a more profound effect than on the ^1^O_2_ generation in HepG2 cells after 24 h of treatment (*P* < 0.05, Figure 8B). These data showed that berberine (**1**), compounds **6e** and **7e** induced apoptosis by increasing the intracellular ROS generation of HepG2 cells

#### 2.2.8. Measurements of Mitochondrial Membrane Potential Changes

The depletion of mitochondrial membrane potential (MMP) is an early marker of the apoptotic process. To determine whether an early loss of MMP occurred upon the treatment of berberine (**1**) and its derivatives in HepG2 cells, we performed MMP measurement with rhodamine 123 staining. Delineated in Figure 9, the MMP decreased remarkably at the concentrations of 0.5 μM of berberine (**1**), 0.25 μM of compound **5e**, **6e**, or **7e**. The percentage of cells with membrane potential loss increased from 20.53% to 32.90% at 24 h. Thus berberine (**1**), compound **5e**, **6e**, or **7e** severely reduced the mitochondrial membrane potential of HepG2 cells. It was documented that P53 could also down-regulate Bcl-2 and up-regulate Bax, which are related to mitochondria-mediated apoptosis [37]. Further, several studies have also reported that apoptosis involves a disruption of the mitochondrial membrane integrity, a critical cell death process. Moreover, the depolarization of the mitochondrial membrane potential is a characteristic feature of apoptosis [38,39]. The report also indicated that the reduction of the membrane potential, the release of mitochondrial cytochrome *c*, a critical step, is followed by the formation of apoptosomes [39]. Thus, we found the loss of MMP upon the treatments of berberine (**1**), compound **5e**, **6e,** or **7e** were consistent with the published results. Thus, does the finding of the loss of mitochondrial potential induced by berberine (**1**), compounds **5e**, **6e**, and **7e** at a concentration of 0.25 μM. These observations suggested that berberine derivatives **6e** and **7e** induced mitochondrial-dependent apoptosis in HepG2 by irradiation.

## 3. Materials and Methods

### 3.1. General Information

All chemical reagents and solvents in commercial quality were purchased from Sigma-Aldrich (St. Louis, MO, USA), Fluka (Roma, Italy), TCI (Tokyo, Japan), and Alpha Aesar without further purification. All chemical reagents in commercial quality were used as received (Sigma-Aldrich, St. Louis, MO, USA) and were used without further purification. Solvents were dried and the synthesized compounds were purified using standard techniques. Reactions-progression was monitored by TLC on aluminum plates coated with silica gel with a fluorescent indicator (Merck 60 F_254_). Melting points of the synthesized compounds were determined by Fargo MP-2D apparatus and were uncorrected. The electronic absorption spectra were measured on a Shimadzu UV-1601 UV-vis spectrophotometer. Fluorescence spectra were taken on a Shimadzu RP-5301 spectrofluorometer. NMR spectra were recorded using TMS as an internal standard in CDCl_3_ at 500 MHz for ^1^H and at 125 MHz for ^13^C (Bruker Biospin GmbH AVANCE III 500 MHz, Rheinstetten, Germany) (see Appendix A). Chemical shift (*δ*) were reported in parts per million (ppm) measured relative to the internal standards (TMS), and the coupling constant (*J*) were expressed in Hertz (Hz). Column chromatography was performed with silica gel Silia*Flash*^®^ G60 (60–200 μm) purchased from SiliCycle Inc. (Quebec City, QC, Canada). In general, the reactions were carried out under anhydrous conditions in dry solvent and nitrogen atmosphere. The purity of these compounds was based on the analysis of HPLC (Hitachi High-Technologies, Tokyo, Japan) equipped with a 280 nm detector and LiChroCART RP-C_18_ column (4.6 mm i.d. × 250 mm, 5 μm, Merck, Darmstadt, Germany). The mobile phase was composed of MeOH-H_2_O (0.05% TFA) (90:10) and the flow rate was 1.0 mL/min. The purity of all compounds was more than 98%. The mass spectra were acquired using a Thermo Finnigan model LXQ (Thermo Electron Co., Waltham, MA, USA) ion trap mass spectrometer equipped with ESI source interference and controlled by Xcalibur 2.06. The mass spectra were acquired in a positive ion mode or a negative ion mode. Berberrubine (**2**), derivatives **3**, **4**, **5a**, **5c,** and **5e** were synthesized according to previous methods [27].

### 3.2. Synthesis of 9-O-Sunstituted Berberine Derivatives (**5a**–**e**)

To a solution of berberrubine **2** (1 mmol) in dry acetonitrile (10 mL) was added various *n*-alkyl bromide (1.2 mmol) and the reaction mixture was refluxed for 4–8 h. The progress of the reaction was monitored by TLC. Then, the reaction mixture was cooled to room temperature and the solvent was removed under reduced pressure. The crude product was chromatographed on a silica gel column and eluted with ethyl acetate/methanol (2/1) solvent to afford the desired berberine derivatives **5a**–**e**.

*9-O-Butylberberine bromide (***5a***).* Berberrubine (**2**) was treated with *n*-butyl bromide according to the general procedure to give the desired bromide salt (**5a**) as a yellowish brown solid, yield: 86%; UV (MeOH) λ_max_ (log *ε*): 431 (3.77), 350 (4.41), 267 (4.43), 231 (4.45), 205 (4.41) nm; ^1^H NMR (500 MHz, DMSO-*d*_6_): *δ* 9.75 (s, 1H, H-8), 8.94 (s, 1H, H-13), 8.20 (d, *J* = 9.1 Hz, H-12), 7.99 (d, *J* = 9.1 Hz, H-11), 7.80 (s, 1H, H-1), 7.09 (s, 1H, H-4), 6.17 (s, 2H, -OCH_2_O-), 4.95 (t, *J* = 6.3 Hz, 2H, H-6), 4.29 (t, *J* = 6.8 Hz, 2H, H-1’), 4.05 (s, 3H, -OCH_3_), 3.21 (t, *J* = 6.3 Hz, 2H, H-5), 1.86 (p, *J* = 7.2 Hz, 2H, H-2’), 1.52 (sex, *J* = 7.4 Hz, 2H, H-3’), 0.99 (t, *J* = 7.4 Hz, 3H, H-4’); ^13^C NMR (125 MHz, DMSO-*d*_6_): *δ* 150.4, 149.8, 147.7, 145.3, 142.9, 137.5, 133.0, 130.7, 126.7, 123.3, 121.7, 120.5, 120.2, 108.4, 105.4, 102.1, 73.9, 57.0, 55.3, 31.5, 26.3, 18.6, 13.8; LC-MS (ESI^+^, *m/z*) C_23_H_24_NO_4_^+^: 378.33 [M–Br]^+^; HRMS-ESI: *m/z* calculated for C_23_H_22_NO_4_^+^: 378.1705 [M − Br]^+^, found for 378.1708.

*9-O-Hexylberberine bromide* (**5b**). Berberrubine (**2**) was treated with *n*-hexyl bromide according to the general procedure to give the desired bromide salt (**5b**) as a yellowish brown solid, yield: 69%; UV (MeOH) λ_max_ (log *ε*): 431.6 (3.92), 350.2 (4.54), 266.5 (4.60), 229.1 (4.67) nm; ^1^H NMR (500 MHz, DMSO-*d*_6_): *δ* 9.75 (s, 1H, H-8), 8.94 (s, 1H, H-13), 8.20 (d, *J* = 9.1 Hz, H-12), 7.99 (d, *J* = 9.1 Hz, H-11), 7.80 (s, 1H, H-1), 7.09 (s, 1H, H-4), 6.17 (s, 2H, -OCH_2_O-), 4.95 (t, *J* = 6.3 Hz, 2H, H-6), 4.28 (t, *J* = 6.8 Hz, 2H, H-1’), 4.05 (s, 3H, -OCH_3_), 3.21 (t, *J* = 6.3 Hz, 2H, H-5), 1.87 (p, *J* = 7.2 Hz, 2H, H-2’), 1.52 (p, *J* = 7.4 Hz, 2H, H-3’),1.37‒1.33 (m, 4H, H-4’, H-5’), 0.90 (t, *J* = 7.0 Hz, 3H, H-6’); ^13^C-NMR (125 MHz, DMSO-*d*_6_): *δ* 150.5, 149.8, 147.7, 145.3, 142.9, 137.5, 133.1, 130.7, 126.7, 123.4, 121.7, 120.5, 120.3, 108.5, 105.5, 102.1, 74.3, 57.1, 55.3, 31.1, 29.5, 26.4, 25.0, 22.1, 14.0; LC-MS (ESI^+^, *m/z*) C_25_H_26_NO_4_^+^: 406.33 [M–Br]^+^; HRMS-ESI: *m/z* calculated for C_25_H_26_NO_4_^+^: 406.2018 [M − Br]^+^, found for 406.2021.

*9-O-Octylberberine bromide* (**5c**). Berberrubine (**2**) was treated with *n*-octyl bromide according to the general procedure to give the desired bromide salt (**5c**) as a bright yellow solid, yield: 74%; UV (MeOH) λ_max_ (log *ε*): 432 (3.79), 350 (4.42), 268 (4.44), 230 (4.47), 204 (4.48) nm; ^1^H NMR (500 MHz, DMSO-*d*_6_): *δ* 9.75 (s, 1H, H-8), 8.95 (s, 1H, H-13), 8.19 (d, *J* = 9.1 Hz, H-12), 7.99 (d, *J* = 9.1 Hz, H-11), 7.80 (s, 1H, H-1), 7.09 (s, 1H, H-4), 6.17 (s, 2H, -OCH_2_O-), 4.95 (t, *J* = 6.2 Hz, 2H, H-6), 4.28 (t, *J* = 6.8 Hz, 2H, H-1’), 4.06 (s, 3H, -OCH_3_), 3.21 (t, *J* = 6.2 Hz, 2H, H-5), 1.87 (p, *J* = 7.2 Hz, 2H, H-2’), 1.48 (p, *J* = 7.3 Hz, 2H, H-3’), 1.33 (m, 8H, H-4’–H-7’), 0.88 (t, *J* = 6.8 Hz, 3H, H-8’); ^13^C NMR (125 MHz, DMSO-*d*_6_): *δ* 150.4, 149.8, 147.7, 145.3, 142.9, 137.5, 133.0, 126.7, 123.3, 121.7, 120.5, 120.2, 108.4, 105.4, 102.1, 74.2, 57.0, 55.3, 31.2, 29.5, 28.8, 28.7, 26.3, 25.3, 22.1, 14.0; LC-MS (ESI^+^, *m/z*) C_27_H_32_NO_4_^+^: 434.41 [M–Br]^+^; HRMS-ESI: *m/z* calculated for C_27_H_30_NO_4_^+^: 434.2331 [M − Br]^+^, found for 434.2332.

*9-O-Decylberberine bromide* (**5d**). Berberrubine (**2**) was treated with *n*-decyl bromide according to the general procedure to give the desired bromide salt (**5d**) as a bright yellow solid, yield: 71%; UV (MeOH) λ_max_ (log *ε*): 431.0 (3.68), 350.2 (4.31), 266.4 (4.35), 229.8 (4.38) nm; ^1^H-NMR (500 MHz, DMSO-*d*_6_): *δ* 9.75 (s, 1H, H-8), 8.95 (s, 1H, H-13), 8.19 (d, *J* = 9.1 Hz, H-12), 7.99 (d, *J* = 9.1 Hz, H-11), 7.80 (s, 1H, H-1), 7.09 (s, 1H, H-4), 6.17 (s, 2H, -OCH_2_O-), 4.95 (t, *J* = 6.2 Hz, 2H, H-6), 4.27 (t, *J* = 6.8 Hz, 2H, H-1’), 4.05 (s, 3H, -OCH_3_), 3.21 (t, *J* = 6.2 Hz, 2H, H-5), 1.87 (p, *J* = 7.2 Hz, 2H, H-2’), 1.48 (p, *J* = 7.3 Hz, 2H, H-3’), 1.40‒1.26 (m, 12H, H-4’–H-9’), 0.85 (t, *J* = 6.8 Hz, 3H, H-10’); ^13^C-NMR (125 MHz, DMSO-*d*_6_): *δ* 150.4, 149.8, 147.7, 145.3, 142.9, 137.5, 133.0, 130.7, 126.7, 123.3, 121.7, 120.5, 120.2, 108.4, 105.4, 102.1, 74.2, 57.0, 55.3, 31.3, 29.5, 29.0, 28.9, 28.8, 28.7, 26.3, 25.3, 22.1, 14.0; LC-MS (ESI^+^, *m/z*) C_29_H_34_NO_4_^+^: 462.43 [M–Br]^+^; HRMS-ESI: *m/z* calculated for C_29_H_34_NO_4_^+^: 462.2644 [M − Br]^+^, found for 462.2646.

*9-O-Dodecylberberine bromide* (**5e**). Berberrubine (**2**) was treated with *n*-dodecyl bromide according to the general procedure to give the desired bromide salt (**5e**) as a dark yellowish brown solid, yield: 61%; UV (MeOH) λ_max_ (log *ε*): 431 (3.42), 351 (4.03), 267 (4.07), 230 (4.10), 202 (4.19) nm; ^1^H NMR (500 MHz, DMSO-*d*_6_): *δ* 9.77 (s, 1H, H-8), 8.98 (s, 1H, H-13), 8.21 (d, *J* = 9.1 Hz, H-12), 8.00 (d, *J* = 9.1 Hz, H-11), 7.82 (s, 1H, H-1), 7.10 (s, 1H, H-4), 6.18 (s, 2H, -OCH_2_O-), 4.96 (t, *J* = 6.1 Hz, 2H, H-6), 4.28 (t, *J* = 6.8 Hz, 2H, H-1’), 4.05 (s, 3H, -OCH_3_), 3.21 (t, *J* = 6.1 Hz, 2H, H-5), 1.88 (d, *J* = 7.4 Hz, H-2’), 1.47 (q, *J* = 7.5 Hz, 2H, H-3’), 1.29 (m, 16 H, H-4’–H-11’), 0.86 (t, *J* = 6.8 Hz, 3H, H-12’); ^13^C NMR (125 MHz, DMSO-*d*_6_): *δ* 150.4, 149.8, 147.7, 145.3, 142.9, 137.5, 133.0, 130.7, 126.7, 123.3, 121.7, 120.5, 120.3, 108.4, 105.5, 102.1, 74.2, 57.0, 55.3, 48.6, 31.3, 29.5, 29.08, 29.05, 28.9, 28.7, 26.3, 25.3, 22.1, 14.0; LC-MS (ESI^+^, *m/z*) C_31_H_38_NO_4_^+^: 490.50 [M − Br]^+^; HRMS-ESI: *m/z* calculated for C_31_H_38_NO_4_^+^: 490.2957 [M − Br]^+^, found for 490.2958.

### 3.3. Synthesis of 13-Sunstituted Berberine Derivatives (**6a**–**e**)

To a stirred solution of berberine (**1**) (3.71 g, 10 mmol) and K_2_CO_3_ (3.6 g, 30 mmol) in methanol (125 mL), 5% NaOH (5 mL) solution containing NaBH_4_ (0.30 g, 7.5 mmol) was added dropwise. The reaction mixture was stirred at room temperature for 1 h and the precipitated product was filtered, washed with 30% ethanol (20 mL) and 80% ethanol (20 mL), and then recrystallized from absolute ethanol to provide dihydroberberine (**3**) (2.5 g, 74%) as a brown like solid, yield: 74%; UV (MeOH) λ_max_ (log *ε*): 353.9 (3.96), 268 (4.01), 228.6 (4.09) nm;^1^H-NMR (500 MHz, DMSO-*d*_6_) *δ* 7.29 (s, 1H, H-13), 6.82 (d, *J* = 8.4 Hz, H-12), 6.76 (s, 1H, H-1), 6.69 (d, *J* = 8.4 Hz, 1H, H-11), 6.05 (s,1H, H-1), 6.00 (s, 2H, -OCH_2_-), 4.20 (s, 2H, H-8), 3.76 (s, 3H, -OCH_3_), 3.71 (s, 3H, -OCH_3_), 3.05 (t, *J* = 5.8 Hz, 2H, H-6), 2.79 (t, *J* = 5.8 Hz, 2H, H-5); ^13^C-NMR (125 MHz, DMSO-*d*_6_) *δ* 149.9, 146.9, 146.4, 143.9, 141.0, 128.6, 128.3, 123.9, 121.6, 118.4, 111.7, 107.9, 103.5, 101.0, 96.0, 60.2, 55.7, 48.8, 48.3, 28.9.

To a stirred solution of dihydroberberine (**3**) (170 mg, 5.0 mmol) in 80% ethanol (8 mL) and HOAc (2 mL), *n*-alkyl aldehyde (2 mL) was added. The reaction mixture was heated to 85–95 °C for 5 h. The solvent was removed by evaporation, and the residue was acidified by 2 N HCl (5 mL), and then stirred at room temperature for 1 h. The solid was collected by filtration and then purified by flash chromatography over silica gel, affording the title compounds **6a**–**e**.

*13-Butylberberine bromide* (**6a**). Dihydroberberine (**3**) was treated with butanal according to the general procedure to give the desired chloride salt (**6a**) as a yellowish brown solid, yield: 68%; UV (MeOH) λ_max_ (log *ε*): 420.3 (3.93), 341.9 (4.49), 265.1 (4.61), 231.9 (4.58) nm; ^1^H-NMR (500 MHz, DMSO-*d*_6_): *δ* 9.88 (s, 1H, H-8), 8.21 (d, *J* = 9.5 Hz, 1H, H-12), 8.18 (d, *J* = 9.5 Hz, 1H, H-11), 7.28 (s, 1H, H-1), 7.16 (s, 1H, H-4), 6.18 (s, 2H, -OCH_2_O-), 4.78 (t, *J* = 8.0 Hz, 2H, H-6), 4.09 (s, 3H, -OCH_3_), 4.08 (s, 3H, -OCH_3_), 3.33 (t, *J* = 8.0 Hz, 2H, H-5), 3.07 (t, *J* = 6.2 Hz, 2H, H-1’), 1.75 (p, *J* = 6.6 Hz, 2H, H-2’), 1.41 (hex, *J* = 7.2 Hz, 2H, H-3’), 0.91 (t, *J* = 6.9 Hz, 3H, H-4’); ^13^C-NMR (125 MHz, DMSO-*d*_6_): *δ* 150.3, 149.1, 146.6, 144.4, 144.3, 135.8, 134.2, 134.1, 132.3, 125.9, 121.5, 121.3, 109.2, 108.4, 102.2, 62.1, 57.1, 57.0, 32.6, 28.8, 27.4, 22.0, 13.5; LC-MS (ESI^+^, *m/z*) C_24_H_26_NO_4_^+^: 392.28 [M − Cl]^+^; HRMS-ESI: *m/z* calculated for C_24_H_26_NO_4_^+^: 392.1862 [M − Cl]^+^, found for 392.1861.

*13-Hexylberberine bromide* (**6b**). Dihydroberberine (**3**) was treated with hexanal according to the general procedure to give the desired chloride salt (**6b**) as a bright yellow solid, yield: 76%; UV (MeOH) λ_max_ (log *ε*): 420.3 (3.73), 341.9 (4.29), 265.4 (4.40), 232 (4.37) nm; ^1^H-NMR (500 MHz, DMSO-*d*_6_): *δ* 9.91 (s, 1H, H-8), 8.22 (d, *J* = 9.7 Hz, 1H, H-12), 8.20 (d, *J* = 9.7 Hz, 1H, H-11), 7.30 (s, 1H, H-1), 7.18 (s, 1H, H-4), 6.20 (s, 2H, -OCH_2_O-), 4.80 (s, 2H, H-6), 4.10 (s, 3H, -OCH_3_), 4.10 (s, 3H, -OCH_3_), 3.33 (s,2H, H-5), 3.09 (t, *J* = 5.7 Hz, 2H, H-1’), 1.76 (s, 2H, H-2’), 1.40 (m, 2H, H-3’), 1.27‒1.26 (m, 4H, H-4’, H-5’), 0.86 (t, *J* = 6.9 Hz, 3H, H-6’); ^13^C-NMR (125 MHz, DMSO-*d*_6_): *δ* 150.2, 149.1, 146.6, 144.4, 144.2, 135.8, 134.2, 134.1, 132.2, 125.9, 121.5, 121.3, 120.3, 109.1, 108.4, 102.2, 62.0, 57.0, 30.7, 30.4, 28.9, 28.3, 27.4, 22.1, 13.9; LC-MS (ESI^+^, *m/z*) C_26_H_30_NO_4_^+^: 420.26 [M − Cl]^+^; HRMS-ESI: *m/z* calculated for C_26_H_30_NO_4_^+^: 420.2175 [M‒Cl]^+^, found for 420.2176.

*13-Octylberberine chloride* (**6c**). Dihydroberberine (**3**) was treated with octanal according to the general procedure to give the desired chloride salt (**6c**) as a bright yellow solid, yield: 65%; UV (MeOH) λ_max_ (log *ε*): 420.1 (3.80), 341.4 (4.35), 265.3 (4.47), 232 (4.44) nm; ^1^H-NMR (500 MHz, DMSO-*d*_6_): *δ* 9.91 (s, 1H, H-8), 8.22 (d, *J* = 9.7 Hz, 1H, H-12), 8.20 (d, *J* = 9.7 Hz, 1H, H-11), 7.30 (s, 1H, H-1), 7.17 (s, 1H, H-4), 6.20 (s, 2H, -OCH_2_O-), 4.80 (s, 2H, H-6), 4.10 (s, 3H, -OCH_3_), 4.10 (s, 3H, -OCH_3_), 3.33 (s,2H, H-5), 3.09 (t, *J* = 5.7 Hz, 2H, H-1’), 1.76 (s, 2H, H-2’), 1.38‒1.37 (m, 2H, H-3’), 1.28‒1.23 (m, 8H, H-4’‒H-7’), 0.86 (t, *J* = 6.9 Hz, 3H, H-8’); ^13^C NMR (125 MHz, DMSO-*d*_6_): *δ* 150.2, 149.0, 146.6, 144.3, 144.2, 135.8, 134.2, 134.0, 132.2, 125.9, 121.4, 121.2, 120.3, 109.1, 108.3, 102.1, 62.0, 57.0, 31.2, 30.4, 28.8, 28.6, 28.4, 27.4, 22.0, 13.9; LC-MS (ESI^+^, *m/z*) C_28_H_34_NO_4_^+^: 448.28 [M–Cl]^+^; HRMS-ESI: *m/z* calculated for C_28_H_34_NO_4_^+^: 448.2488 [M − Cl]^+^, found for 448.2487.

*13-Decylberberine chloride* (**6d**). Dihydroberberine (**3**) was treated with decanal according to the general procedure to give the desired chloride salt (**6d**) as a bright yellow solid, yield: 43%; UV (MeOH) λ_max_ (log *ε*): 420.3 (3.73), 341.9 (4.29), 265.3 (4.40), 231.7 (4.39) nm; ^1^H-NMR (500 MHz, DMSO-*d*_6_): *δ* 9.90 (s, 1H, H-8), 8.22 (d, *J* = 9.7 Hz, 1H, H-12), 8.20 (d, *J* = 9.7 Hz, 1H, H-11), 7.29 (s, 1H, H-1), 7.16 (s, 1H, H-4), 6.19 (s, 2H, -OCH_2_O-), 4.79 (s, 2H, H-6), 4.09 (s, 3H, -OCH_3_), 4.09 (s, 3H, -OCH_3_), 3.33 (s, 2H, H-5), 3.08 (t, *J* = 5.7 Hz, 2H, H-1’), 1.74 (s, 2H, H-2’), 1.36‒1.35 (m, 2H, H-3’), 1.27‒1.22 (m, 12H, H-4’‒H-9’), 0.86 (t, *J* = 6.9 Hz, 3H, H-10’); ^13^C-NMR (125 MHz, DMSO-*d*_6_): *δ* 150.2, 149.0, 146.6, 144.3, 144.2, 135.8, 134.2, 134.0, 132.2, 125.9, 121.4, 121.2, 120.3, 109.1, 108.3, 102.1, 62.0, 57.0, 31.3, 30.4, 28.9, 28.8, 28.6, 28.8, 28.4, 27.4, 22.1, 13.9; LC-MS (ESI^+^, *m/z*) C_30_H_38_NO_4_^+^: 476.40 [M–Cl]^+^; HRMS-ESI: *m/z* calculated for C_30_H_38_NO_4_^+^: 476.2801 [M − Cl]^+^, found for 476.2801.

*13-Dodecylberberine chloride* (**6e**). Dihydroberberine (**3**) was treated with dodecanal according to the general procedure to give the desired chloride salt (**6e**) as a bright yellow solid, yield: 29%; UV (MeOH) λ_max_ (log *ε*): 420.6 (3.71), 341.8 (4.26), 265.3 (4.38), 232.0 (4.37) nm; ^1^H-NMR (500 MHz, DMSO-*d*_6_): *δ* 9.90 (s, 1H, H-8), 8.21 (d, *J* = 9.6 Hz, 1H, H-12), 8.19 (d, *J* = 9.6 Hz, 1H, H-11), 7.29 (s, 1H, H-1), 7.16 (s, 1H, H-4), 6.18 (s, 2H, -OCH_2_O-), 4.79 (s, 2H, H-6), 4.09 (s, 3H, -OCH_3_), 4.09 (s, 3H, -OCH_3_), 3.33 (s,2H, H-5), 3.08 (t, *J* = 5.7 Hz, 2H, H-1’), 1.74 (s, 2H, H-2’), 1.36‒1.35 (m, 2H, H-3’), 1.28‒1.22 (m, 16H, H-4’‒H-11’), 0.85 (t, *J* = 6.9 Hz, 3H, H-12’); ^13^C-NMR (125 MHz, DMSO-*d*_6_): *δ* 150.2, 149.0, 146.5, 144.3, 144.2, 135.8, 134.2, 134.0, 132.2, 125.9, 121.4, 121.2, 120.3, 109.1, 108.3, 102.1, 62.0, 57.0, 31.3, 30.4, 30.0, 28.9, 28.9, 28.8, 28.7, 28.6, 28.4, 27.4, 22.1, 13.9; LC-MS (ESI^+^, *m/z*) C_32_H_42_NO_4_^+^: 504.43 [M − Cl]^+^; HRMS-ESI: *m/z* calculated for C_32_H_42_NO_4_^+^: 504.3114 [M − Cl]^+^, found for 504.3115.

### 3.4. Synthesis of 13-O-Sunstituted Berberine Derivatives (**7a**–**e**)

Compound **3** (337 mg) was dissolved in 35 mL of CH_2_Cl_2_. Under nitrogen atmosphere, the temperature of the system was kept at ‒25 °C ~ ‒30 °C, and 8 mL of CH_2_Cl_2_ solution containing 258 mg MCPBA (1.5 mmol) was added slowly dropwise. After the addition, the temperature was kept and the reaction was carried out under agitation for 1 h. Then, the temperature was raised to 0 °C and 250 mg (2.0 mmol) of sodium sulfite was added therein, followed by being stirred at room temperature for 1 h. The reaction was stopped and the reaction mixture was filtered. The filtrate was evaporated to remove the solvent under reduced pressure and the residue was purified through silica gel column chromatography (CH_2_Cl_2_/CH_3_OH = 10:1) to obtain compound **4** (280 mg, 80 percent), yield: 80%;UV (MeOH) λ_max_ (log *ε*): 444.5 (4.08), 367.9 (3.93), 305.3 (4.04), 235.0 (4.44), 212.1 (4.42) nm; ^1^H-NMR (500 MHz, DMSO-*d*_6_): *δ* 8.89 (s, 1H, H-8), 8.08 (d, *J* = 9.3 Hz, 1H, H-12), 8.06 (s, 1H, H-1), 7.52 (d, *J* = 9.3 Hz, 1H, H-11), 6.88 (s, 1H, H-4), 6.02 (s, 2H, -OCH_2_O-), 4.61 (t, *J* = 6.2 Hz, 2H, H-6), 3.95 (s, 3H, -OCH_3_), 3.93 (s, 3H, -OCH_3_), 3.01 (t, *J* = 6.2 Hz, 2H, H-5); ^13^C-NMR (125 MHz, DMSO-*d*_6_): *δ* 165.1, 150.2, 145.3, 145.2, 142.2, 130.2, 127.4, 123.9, 123.4, 122.3, 121.4, 117.6, 117.3, 107.5, 107.2, 100.8, 61.1, 56.4, 56.3, 27.6.

Compound **4** (1.0 g) and potassium carbonate (200 mg) were dissolved in 20 mL of acetone followed by adding 0.25 mL of alkyl bromide. The obtained reaction solution was heated and refluxed for 3 h. Then, the reaction mixture was filtered and the filtrate was evaporated under reduced pressure. The residual was purified through silica gel column (CH_2_Cl_2_/CH_3_OH = 20: 1) to obtain compounds **7a**–**e**.

*13-O-Butylberberine bromide* (**7a**). Compound (**4**) was treated with *n*-butyl bromide according to the general procedure to give the desired bromide salt (**7a**) as a yellowish brown solid, yield: 66%; UV (MeOH) λ_max_ (log *ε*): 427.8 (3.46), 349.7 (3.97), 265.5 (4.07), 232.6 (4.17) nm;^1^H-NMR (500 MHz, DMSO-*d*_6_): *δ* 9.82 (s, 1H, H-8), 8.21 (d, *J* = 9.3 Hz, 1H, H-12), 8.09 (d, *J* = 9.3 Hz, 1H, H-11), 7.90 (s, 1H, H-1), 7.13 (s, 1H, H-4), 6.17 (s, 2H, -OCH_2_O-), 4.88 (t, *J* = 5.8 Hz, 2H, H-6), 4.10 (s, 3H, -OCH_3_), 4.09 (s, 3H, -OCH_3_), 3.84 (t, *J* = 6.4 Hz, 2H, H-1’), 3.17 (t, *J* = 4.7 Hz, 2H,H-5), 1.80‒1.74 (m, 2H, H-2’), 1.53‒1.45 (m, 2H, H-3’), 0.91 (t, *J* = 7.4 Hz, 3H, H-4’); ^13^C-NMR (125 MHz, DMSO-*d*_6_): *δ* 150.9, 150.0, 149.2, 146.8, 144.2, 142.0, 132.5, 130.8, 128.8, 126.1, 122.1, 118.4, 108.4, 107.9, 102.1, 75.0, 62.0, 57.1, 56.4, 31.5, 26.9, 18.6, 13.6; HRMS-ESI: *m/z* calculated for C_24_H_26_NO_5_^+^: 408.1811 [M − Br]^+^, found for 408.1812.

*13-O-Hexylberberine bromide* (**7b**). Compound (**4**) was treated with *n*-hexyl bromide according to the general procedure to give the desired bromide salt (**7b**) as a bright yellow solid, yield: 76%; UV (MeOH) λ_max_ (log *ε*): 428.0 (3.75), 350.3 (4.26), 266.5 (4.32), 229.0 (4.45) nm; ^1^H-NMR (500 MHz, DMSO-*d*_6_): *δ* 9.81 (s, 1H, H-8), 8.21 (d, *J* = 9.4 Hz, 1H, H-12), 8.09 (d, *J* = 9.4 Hz, 1H, H-11), 7.90 (s, 1H, H-1), 7.13 (s, 1H, H-4), 6.17 (s, 2H, -OCH_2_O-), 4.87 (t, *J* = 5.9 Hz, 2H, H-6), 4.10 (s, 3H, -OCH_3_), 4.09 (s, 3H, -OCH_3_), 3.84 (t, *J* = 6.4 Hz, 2H, H-1’), 3.16 (t, *J* = 5.9 Hz, 2H, H-5), 1.78 (p, *J* = 7.0 Hz, 2H, H-2’), 1.45 (p, *J* = 7.5 Hz,2H, H-3’), 1.33‒1.23 (m, 4H, H-4’, H-5’), 0.86 (t, *J* = 6.9 Hz, 3H, H-6’); ^13^C-NMR (125 MHz, DMSO-*d*_6_): *δ* 150.9, 150.1, 149.2, 146.8, 144.2, 141.9, 132.5, 130.8, 128.8, 126.1, 122.1, 118.5, 118.4, 108.3, 108.0, 102.1, 75.4, 62.0, 57.1, 30.9, 29.4, 25.1, 22.0, 13.9; HRMS-ESI: *m/z* calculated for C_26_H_30_NO_5_^+^: 436.2124 [M − Br]^+^, found for 436.2122.

*13-O-Octylberberine chloride* (**7c).** Compound (**4**) was treated with *n*-octyl bromide according to the general procedure to give the desired bromide salt (**7c**) as a bright yellow solid, yield: 61%; UV (MeOH) λ_max_ (log *ε*): 430.6 (3.73), 349.2 (4.23), 265.7 (4.32), 234.0 (4.42) nm; ^1^H-NMR (500 MHz, CDCl_3_): *δ* 10.33 (s, 1H, H-8), 7.98 (d, *J* = 9.2 Hz, 1H, H-12), 7.94 (s, 1H, H-1), 7.83 (d, *J* = 9.2 Hz, 1H, H-11), 6.85 (s, 1H, H-4), 6.09 (s, 2H, -OCH_2_O-), 5.24 (d, *J* = 5.8 Hz, 1H, H-6), 4.34 (s, 3H, -OCH_3_), 4.08 (s, 3H, -OCH_3_), 3.83 (d, *J* = 6.5 Hz, 2H, H-1’), 3.29 (t, *J* = 5.8 Hz, 2H, H-5), 1.83 (p, *J* = 7.1 Hz, 2H, H-2’), 1.50 (p, *J* = 7.2 Hz, 2H, H-3’), 1.33‒1.26 (m, 8H, H-4’‒H-7’), 0.90 (t, *J* = 6.8 Hz, 3H, H-8’); ^13^C NMR (125 MHz, CDCl_3_): *δ* 152.3, 150.7, 149.9, 147.1, 146.2, 143.4, 132.1, 130.9, 129.9, 125.4, 123.0, 118.5, 117.9, 108.5, 108.4, 102.0, 75.9, 63.2, 57.1, 31.8, 30.2, 29.3,29.2, 28.2, 26.0, 22.6, 14.1; LC-MS (ESI^+^, *m/z*) C_28_H_34_NO_5_^+^: 464.58 [M − Br]^+^; HRMS-ESI: *m/z* calculated for C_28_H_34_NO_5_^+^: 464.2437 [M‒Br]^+^, found for 464.2436.

*13-O-Decylberberine chloride* (**7d**). Compound (**4**) was treated with *n*-decyl bromide according to the general procedure to give the desired bromide salt (**7d**) as a bright yellow solid, yield: 58%; UV (MeOH) λ_max_ (log *ε*): 428.3 (3.70), 349.6 (4.19), 265.4 (4.28), 223.8 (4.47) nm; ^1^H-NMR (500 MHz, CDCl_3_): *δ* 10.50 (s, 1H, H-8), 7.97 (d, *J* = 9.2 Hz, 1H, H-12), 7.94 (s, 1H, H-1), 7.81 (d, *J* = 9.2 Hz, 1H, H-11), 6.86 (s, 1H, H-4), 6.09 (s, 2H, -OCH_2_O-), 5.31 (d, *J* = 5.8 Hz, 1H, H-6), 4.34 (s, 3H, -OCH_3_), 4.09 (s, 3H, -OCH_3_), 3.82 (d, *J* = 6.5 Hz, 2H, H-1’), 3.28 (t, *J* = 5.8 Hz, 2H, H-5), 1.83 (p, *J* = 7.2 Hz, 2H, H-2’), 1.50 (d, *J* = 7.2 Hz, 2H, H-3’), 1.32‒1.23 (m, 12H, H-4’‒H-9’), 0.90 (t, *J* = 6.9 Hz, 3H, H-10’); ^13^C-NMR (125 MHz, CDCl_3_): *δ* 151.3, 150.6, 149.9, 147.5, 146.4, 143.9, 132.2, 130.8, 129.8, 125.3, 123.1, 118.5, 117.8, 108.4, 102.0, 75.8, 63.2, 57.0, 56.7, 31.9, 30.1,29.5, 29.33, 29.29, 28.2, 26.0, 22.7, 14.1; LC-MS (ESI^+^, *m/z*) C_30_H_38_NO_5_^+^: 492.62 [M − Br]^+^; HRMS-ESI: *m/z* calculated for C_30_H_38_NO_5_^+^: 492.2750 [M − Br]^+^, found for 492.2751.

*13-O-Dodecylberberine chloride* (**7e**). Compound (**4**) was treated with *n*-dodecyl bromide according to the general procedure to give the desired bromide salt (**7e**) as a bright yellow solid, yield: 40%; UV (MeOH) λ_max_ (log *ε*): 427.5 (3.63), 350 (4.13), 266.4 (4.20), 225.4 (4.37) nm; ^1^H-NMR (500 MHz, DMSO-*d*_6_): *δ* 9.80 (s, 1H, H-8), 8.20 (d, *J* = 9.4 Hz, 1H, H-12), 8.09 (d, *J* = 9.4 Hz, 1H, H-11), 7.89 (s, 1H, H-1), 7.13 (s, 1H, H-4), 6.16 (s, 2H, -OCH_2_O-), 4.86 (t, *J* = 5.9 Hz, 2H, H-6), 4.09 (s, 3H, -OCH_3_), 4.08 (s, 3H, -OCH_3_), 3.84 (t, *J* = 6.4 Hz, 2H, H-1’), 3.16 (t, *J* = 5.9 Hz, 2H, H-5),1.77 (p, *J* = 7.0 Hz, 2H, H-2’), 1.43(m, 2H, H-3’), 1.24 (m, 16H, H-4’‒H-11’), 0.85 (t, *J* = 6.9 Hz, 3H, H-12’); ^13^C-NMR (125 MHz, DMSO-*d*_6_): *δ* 150.9, 150.1, 149.2, 146.8, 144.2, 142.0, 132.5, 130.9, 128.8, 126.1, 122.2, 118.5, 118.4, 108.3, 108.0, 102.1, 75.4, 62.0, 57.1, 56.4, 31.3, 29.4, 29.04, 29.02, 28.96, 28.89, 28.75, 28.7, 26.9, 25.4, 22.1, 14.0; LC-MS (ESI^+^, *m/z*) C_32_H_42_NO_5_^+^: 520.68 [M‒Br]^+^; HRMS-ESI: *m/z* calculated for C_32_H_42_NO_5_^+^: 520.3063 [M − Br]^+^, found for 520.3066.

### 3.5. Biological Activity

#### 3.5.1. Cell Lines and Cell Culture Conditions

All cells were obtained from the Bioresource Collection and Research Center (Hsinchu, Taiwan). Human colon cancer cells (HT-29) was cultured in RPMI 1640 (HyClone, South Logan, UT, USA) medium, supplemented with 10% fetal bovine serum (FBS, Gibco, Life Technologies, Carlsbad, CA, USA) and antibiotics (100 mg/mL streptomycin and 100 U/mL penicillin, Gibco, Life Technologies, USA). Human bladder cancer cells (BFTC 905) and human liver carcinoma (HepG2) cell lines were cultured in Dulbecco’s modified Eagle medium (DMEM, HyClone, USA) medium, supplemented with 10% fetal bovine serum (FBS) and antibiotics. Human cancer cells were maintained in humidified atmosphere with 5% CO_2_ and 95% air at 37 °C in a carbon dioxide incubator (SANYO, CO_2_ incubator, Osaka, Japan).

#### 3.5.2. Cell Viability Inhibition Assay

Cell viability was assessed by staining with 3-(4,5-dimethylthiazol-2-yl)-2,5-diphenyl tetrazolium bromide (MTT, Bionovas Biotechnology Co., Ltd., Toronto, Ontario, Canada) [32]. Briefly, the cancer cells were plated at 5 × 10^3^ – 1 × 10^4^ cells into 96-well plates incubated at 37 °C in a humidified 5% CO2 incubator for 24 h. After medium removal, 100 μL of culture medium with 0.1% DMSO containing the test compounds at different concentrations was added to each well and incubated at 37 °C for another 24 h. At the end of treatment, final concentration of 0.5 mg/mL MTT was added, and cells were incubated for a further 1.5 h. The absorbance was recorded on an ELISA plate reader (SpectraMax 340PC^384^, Molecular Devices, Orleans, CA, USA) at a 540 nm wavelength. Inhibition ratio (%) was calculated using the following equation:Inhibition ratio (%) = [(A_control_ − A_treated_)/A_control_] × 100%,(1)
A_treated_ and A_control_ are the average absorbance of three independent experiments from treated and control groups, respectively.

Cell viability was determined the test compound concentration required to inhibit tumor cell proliferation by 50% (IC_50_) from the dose-response curves. All data average values from triplicate samples and the experiments were repeated at least three times.

#### 3.5.3. Measurement of Absorption and Fluorescence Spectra

Absorption spectrum (250–500 nm) of berberine and its derivatives (**5e**, **6e,** and **7e**) were recorded on a UV–vis spectrophotometer (UV-1601, Shimadzu, Kyoto, Japan) with a quartz cuvette (path length 1 cm, accuracy ± 0.2 nm). Fluorescence emission spectrum were recorded on a fluorescence spectrophotometer (RF-5301PC, Shimadzu, Kyoto, Japan) with excitation at 420 nm and a measure of emission from 450 to 700 nm (slit 3/3 nm) in a 1 cm quartz cuvette. A final concentration of berberine and its derivatives **5e**, **6e,** and **7e** were 10 μM diluted using methanol. Appropriate blanks corresponding to the methanol were subtracted to correct the background.

#### 3.5.4. Light Source

The UVA light box consisting of 8 UVA lamps (Blacklight blue, F8T5BLB, Sankyo Denki Co. Ltd., Osaka, Japan) was custom made [37]. The light spectral irradiance of the light box was determined using a NIST-Traceable radiometer (ILT1400 Portable radiometer, International Light Technologies Inc., MA, USA). The light dose of the UVA light box, with the maximum emission at 420 nm, was routinely measured using a photochemical reactor (PR-1000, Panchum Scientific Corp., Kaohsiung, Taiwan). In the study, the light-irradiation doses were determined at 1.8, 3.6, 7.2 J/cm^2^ for 420 nm (dose rate 5.9 mW/cm^2^), which were obtained by approximately 5, 10, and 20 minutes of exposure, respectively. To determine visible light dose using the following equation:*E* (J/cm^2^) = *t* (s) ×*P* (W/cm^2^),(2)
where *E* stands for visible light energy, *t* represents time expressed in second and, finally, *P* is lamp potency.

#### 3.5.5. Photocytotoxic Assay

The photocytotoxicity of berberine (**1**) and its derivatives (**5e**, **6e,** and **7e**) were studied by using the MTT colorimetric assay with quantification of formazan formed by cleavage of the tetrazolium rings of MTT in DMSO by spectral measurement [32]. Various concentrations of the berberine (**1**) and berberine derivatives (**5e**, **6e,** and **7e**) dissolved in 1% DMSO were added to the cells and incubation was continued for 4 h in the dark, followed by photo-irradiation with visible light of 420 nm (1.8, 3.6 and 7.2 J/cm^2^ for 5, 10 and 20 min of irradiation, respectively) by using a photochemical reactor (PR-1000, Panchum Scientific Corp., Kaohsiung, Taiwan) filled with 8 fluorescent black tubes lamps (Blacklight blue, F8T5BLB, Sankyo Denki Co. Ltd., Osaka, Japan). For these experiments, the cells were seeded in 96-well microliter plates (5000 cells/well) and grown for at least 24 h in 100 μL DMEM containing 10% FBS. The growth medium was replaced with fresh medium containing 3% FBS and berberine (5, 10, 20, and 40 μM) and its derivatives (0.05, 0.25, 0.5, and 1.0 μM) and the cells were incubated for 4 h at 37 °C. At the end of the incubation, the drug-containing medium was removed, and the cells were washed with fresh medium and irradiated for 5, 10, and 20 min with visible light (420 nm). After the incubation period for 20 h, 10 μL MTT (5 mg/mL) was added to each well and incubated for an additional 1.5 h at 37 °C, 100 μL DMSO was added to dissolve the formazan crystals. The absorbance of the samples was measured using an ELISA microplate reader (SpectraMax 340PC^384^, Molecular Devices, Orleans, CA, USA) and selecting 540 nm as the lest wavelengths. The survival of the treated cells was calculated as the percentage of their absorbance referred to the absorbance of the untreated (no drug, no light) cells taken as 100% viability. The IC_50_ values were obtained by nonlinear regression analysis (Sigma 11.0 software).

#### 3.5.6. Intracellular Uptake Profiling of Berberine and its Derivatives in HepG2 Cells

In order to establish a time profile for intracellular accumulation of these berberine derivatives, an HPLC analysis was carried out. The HepG2 cells treated with different berberine (**1**) and its derivatives (**5e**, **6e,** and **7e**) were allowed to grow in 5 mL liquid culture. The cultures were incubated in a shaking incubator in the dark. After 0 h, 0.25 h, 1 h, 2 h, 4h, and 8 h, aliquots of cells were harvested by centrifugation at 6000 rpm for 5 min. The supernatant liquid was discarded and the cell pellet was washed thrice with 1 mL of 2×PBS in order to remove the uninternalized berberine and its derivatives. After this, the cells were gently resuspended in 0.2 mL methanol and then were filtered with a 0.22 μm filter into HPLC analysis. The HPLC analysis was carried out by exciting the sample at 350 nm and emission at 530 nm.

#### 3.5.7. Cell Morphological Assessment

To detect morphological evidence of apoptosis, cells were visualized following DNA staining with the fluorescent dye Hoechst 33258 [34]. The HepG2 cells were plated in a 6 cm dish at a density of 7×10^5^ cells/well and were incubated overnight. The cells were treated with 0.5 μM concentrations berberine (**1**), compounds **5e**, **6e,** and **7e** for 4 h in the dark and treated with visible light (420 nm) for 10 min. Additionally, the HepG2 cells of compound **7e** treatment were stained with 10 μM Hoechst 33258 (Sigma-Aldrich, St. Louis, MO, USA). After Cells were incubated in a dark room for 5 min, the cells were washed with PBS and then were observed by a fluorescent microscope (Zeiss, Axio Observer A1, Japan).

#### 3.5.8. Cell Cycle Distribution Analysis

Flow cytometry was used to obtain the cell cycle distribution and the apoptotic rate [35]. The HepG2 cells were plated at 1×10^6^ cells per 6 cm dish and cultured overnight. Then, various concentrations of 0.5 μM berberine (**1**), compounds **5e**, **6e,** and **7e** were added for 2 h. At the end of the incubation, the drug-containing medium was removed, and the cells were washed with fresh medium and irradiated for 10 min with visible light (420 nm). The cells of each dish were harvested, washed once with PBS, and fixed with methanol at 4 °C. After 24 h, the cells of each dish were harvested and washed once with PBS. The cells were suspended in 473 μL of PBS containing 40 μg/mL propidium iodide (PI) and 40 μg/mL RNase. Cells were incubated in a dark room for 30 min at room temperature then subjected to cell cycle analysis using a FACScan flow cytometer (Becton Dickinson, San Jose, CA, USA) and analyzed using the ModFit 3.0 software (Verity Software House, ME, USA).

#### 3.5.9. Measurement of Intracellular Singlet Oxygen Production

Intracellular oxidant stress was monitored by measuring changes in fluorescence resulting from intracellular probe oxidation. Intracellular production of ROS, namely singlet oxygen (^1^O_2_), was measured using the DCFH-DA (Molecular Probes, USA) probe, respectively [36]. The level of intracellular singlet oxygen was detected using fluorescent dye DCF sensitivity. Briefly, the HepG2 (4×10^4^ cells/ml) cells were pretreated with 0.5 μM of berberine (**1**) and 0.25 μM of its derivative (**5e**, **6e** and **7e**) for 2 h. At the end of the incubation, the drug-containing medium was removed, and the cells were washed with fresh medium and irradiated for 10 min with visible light (420 nm). After the incubation period for 24 h, intracellular peroxide level was detected by DCFH-DA as an intracellular fluorescence probe. After incubation with 20 μM DCFH-DA for 1 h, the cells were harvested and washed with PBS twice. A flow cytometer (Becton Dickinson, San Jose, CA, USA) was used to detect dichlorofluorescein (DCF). Data were collected and analyzed by ModFit 3.0 software (Verity Software House, ME, USA).

#### 3.5.10. Measurement of the Mitochondrial Membrane Potential

The changes in the mitochondrial membrane potential (MMP) were determined by using the fluorescent probe rhodamine 123 (Sigma, St Louis, USA). The dye Rh123 selectively enters mitochondria without influencing the membrane potential and is retained inside the mitochondria [37]. Once the MMP is lost, rhodamine 123 is subsequently washed out of the cells, leading to the decline of the fluorescence signal. HepG2 cells (1×10^5^ cells/mL) were incubated with concentrations of 0.5 μM berberine (**1**) and 0.25 μM of its derivative (**5e**, **6e**, and **7e**) for 2 h. At the end of the incubation, the drug-containing medium was removed, and the cells were washed with fresh medium and irradiated for 10 min with visible light (420 nm) for 24 h. Then, stained with 2 mg/mL of rhodamine 123 for 20 min under gentle shaking. After washing with PBS, cells were immediately analyzed by flow cytometer (Becton Dickinson, San Jose, CA, USA) and data were analyzed using the ModFit 3.0 software (Verity Software House, ME, USA).

#### 3.5.11. Statistical Analysis

The values shown are mean ± SD of three independent experiments. Data are statistically evaluated by Student’s *t*-test of SigmaPlot 11.0 and shown significantly different in * *p* < 0.05, ** *p* < 0.01, and *** *p* < 0.001.

## 4. Conclusions

In summary, berberine was mortified with *n*-alkyl groups of different chain lengths and studied for their photo-induced cytotoxicity against three human cancer cell lines. Out of the 15 synthesized derivatives, we found 3; compounds **5e**, **6e**, and **7e,** worked the best in vitro against the cancer cell lines. The increasing chain length of *n*-alkyl groups enhanced lipophilicity and the cellular permeability of the derivatives. Besides, its derivatives, compounds **5e**, **6e**, and **7e** had exhibited low dark toxicity and higher photocytotoxicity toward the cancer cell lines under testing. Thus derivatives **5e**, **6e**, and **7e** with a dodecyl group on the 9-, 13- or 13-*O*-position of berberine, respectively, showed high cellular uptake, photo-induced cytotoxicity in vitro, ROS generation, and induced cell apoptosis after 420 nm visible light irradiation. In the cell cycle distribution and apoptotic analysis, compound **5e**, **6e**, or **7e** induced S phase arrested and apoptosis in HepG2 cells after visible light irradiation. In Hoechst 33,258 staining analysis, these three compounds markedly induce apoptosis, which was further confirmed by the Sub-G1 rate of cell cycle analysis. Taken together, the presented results showed that compounds **5e**, **6e,** and **7e** increased cellular accumulation and enhanced the anti-cancer activity by photo-induced therapy against the cancer cells. We thus believe they could be novel anti-cancer drug candidates.

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
