# Peer review of "Synthesis and In Vitro Photocytotoxicity of 9-/13-Lipophilic Substituted Berberine Derivatives as Potential Anticancer Agents"

_molecules, 2020, doi:10.3390/molecules25030677_

Round 1

Reviewer 1 Report

The manuscript „Synthesis and in vitro photocytotoxicity of 9-/13-lipophilic substituted berberine derivatives as potential anticancer agents” by Lin HJ et al. presents the results of synthesis and antitumor activities of synthesized berberine derivatives toward HepG2, BFTC905 and HT-29 tumour cell lines. Despite the prepared berberine derivatives are not absolutely new and have been already synthesized by other groups, the antitumor and phototoxic activities of these derivatives have not been tested toward selected tumour cell lines. Thus the presented results are interesting.

However, the manuscript contains a lot of discrepancies. The results are not always clearly presented and discussed within the context of already published literature. Additionally, the manuscript needs a round of editing by a native English speaker.

Recommendations:

Lane 32 – berberine derivative 5e did not elevate the ROS (Fig. 8), check the data and correct the sentence

Lanes 56 to 59 – this paragraph might be deleted

Lane 79 - some recent works on very similar molecules should be discussed, and results presented put in context (e.g. PMIDs: 30014806, 31052469, 28325147)

Lane 128 – it is cytotoxicity, not lipophilicity presented in Fig. 2

Lanes 223 to 235 – this part has to be moved to Methods section

Lane 243 – this is not true for berberine derivative 5e

Lane 249 (Table 2) – please try to explain and discuss why the berberine derivative 5e after irradiation is only two times more potent against HepG2 cells compare to other cell lines tested, where it behaves similarly as berberine derivatives 6e and 7e (10-15 times higher cytotoxicity after irradiation)

Lane 259 to 262 – this is not what Fig. 6A is about. Actually, it assesses changes in phase-contrast morphology of treated cells.

Lane 263 and Fig. 6B – it is essential to show Hoechst staining of HepG2 cells treated with all tested berberine derivatives (5e, 6e and 7e) to be able to declare that berberine derivatives induce apoptosis based on these data

Lane 281 – explain the reason you incubated the HepG2 cells with tested berberine derivatives for 2 hours only when testing their effects on cell cycle distribution, versus 4 hours in the case of phototoxicity testing, check lane 226

Lane 302, Fig. 7A – explain the link between cytotoxicity and photocytotoxicity of berberine derivative 5e, as it is more potent at the tested concentrations even without irradiation, but does not induce apoptosis. The cytotoxicity and photocytotoxicity is quite similar, isn't it?  What could be a reason?

Lane 316 – this is not true for berberine derivative 5e. Based on the results of Fig. 8, there is no increase in the level of ROS in HepG2 cells treated with berberine derivative 5e.

Lane 336 – explain a reason for decreasing concentrations of berberine derivatives when measuring their effect on mitochondrial membrane potential

Lane 338 – should be “mitochondrial potential” not “damaged the DNA”

Lane 338 to 344 – this paragraph has to be discussed within the context of apoptosis inducing activities of previously published berberine derivatives

Lane 345 – results on cytochrome c release from mitochondria are not presented in the manuscript. Include the data, or rewrite the sentence.

Concerning the synthesis of berberine derivatives, authors have to be more specific and specify exact time for each reaction steps, including the reflux (e.g. lane 387).

Lane 724 – there are no V-FITC/PI data presented in this manuscript

Author Response

Reviewer 1-comments:

The manuscript „Synthesis and in vitro photocytotoxicity of 9-/13-lipophilic substituted berberine derivatives as potential anticancer agents” by Lin HJ et al. presents the results of synthesis and antitumor activities of synthesized berberine derivatives toward HepG2, BFTC905 and HT-29 tumour cell lines. Despite the prepared berberine derivatives are not absolutely new and have been already synthesized by other groups, the antitumor and phototoxic activities of these derivatives have not been tested toward selected tumour cell lines. Thus the presented results are interesting.

However, the manuscript contains a lot of discrepancies. The results are not always clearly presented and discussed within the context of already published literature. Additionally, the manuscript needs a round of editing by a native English speaker.

Recommendations:

Lane 32 – berberine derivative 5e did not elevate the ROS (Fig. 8), check the data and correct the sentence

Reply comment:

Thank you so very much for your recommendations. 

The phototoxic effects and general cytotoxicity led us conclude that compound 5e has both ROS-related pathways and mitochondrial-related pathways of apoptosis.

Lanes 56 to 59 – this paragraph might be deleted

Reply comment:

It was deleted in the revised manuscript.

Lane 79 - some recent works on very similar molecules should be discussed, and results presented put in context (e.g. PMIDs: 30014806, 31052469, 28325147)

Reply comment:

Three references were added to the manuscript.

The text was amended: ”Study have also showed that the introduction of a lipophilic substituent into 9-O-position of berberine backbone enhance the anti-proliferative activities against human cancer cell lines [27-30].” in the revised manuscript.

Xiao, D.; He, F.; Peng, D.; Zou, M.; Peng, J.; Liu, P.; Liu, Y.; Liu, Z. Synthesis and anticancer activity of 9-O-pyrazole alkyl substituted berberine derivatives. Anticancer Agents Med. Chem. 2018, 18, 1639–1648. Milata, V.; Svedova, A.; Barbierikova, Z.; Holubkova, E.; Cipakova, I.; Cholujova, D.; Jakubikova, J.; Panik, M.; Jantova, S.; Brezova, V.; Cipak, L. Synthesis and anticancer activity of novel 9-O-substituted berberine derivatives. Int. J. Mol. Sci. 2019, 20, 2169. Xiao, D.; Liu, Z.; Zhang, S.; Zhou, M.; He, F.; Zou, M.; Peng, J.; Xie, X.; Liu, Y.; Peng, D. Berberine derivatives with different pharmacological activities via structural modifications. Mini Rev. Med. Chem. 2018, 18, 1424–1441.

Lane 128 – it is cytotoxicity not lipophilicity presented in Fig. 2

Reply comment:

It was corrected in the revised manuscript.

Lanes 223 to 235 – this part has to be moved to Methods section

Reply comment:

It was revised as suggested.

Lane 243 – this is not true for berberine derivative 5e

Reply comment:

It was corrected in the revised manuscript.

Lane 249 (Table 2) – please try to explain and discuss why the berberine derivative 5e after irradiation is only two times more potent against HepG2 cells compare to other cell lines tested, where it behaves similarly as berberine derivatives 6e and 7e (10-15 times higher cytotoxicity after irradiation)

Reply comment:

We believe that compound 5e has phototoxic effects and dark cytotoxicity.

Lane 259 to 262 – this is not what is Fig. 6A about. Actually, it assesses changes in phase-contrast morphology of treated cells.

Reply comment:

The text was amended: ”As shown in Fig. 6A, while the nuclei of the cells were round in shape homogenously in the control without testing compound, those treated with berberine (1), compound 5e, 6e or 7e showed typical morphological features of apoptosis such as cell blobbing and nuclear shrinkage. Fig. 6B shows that HepG2 cells treated with 0.5μM of compound 7e stained with Hoechst 33258 in dark or light, and examined under fluorescence microscopy for topical morphological changes, such as chromatin condensation and DNA fragmentation [33]. Evidently, the proliferation of the cancer cells was inhibited by the testing compounds in vitro. These results showed that compounds 5e, 6e or 7e induce apoptosis in HepG2 cells through irradiation.”.

Lane 263 and Fig. 6B – it is essential to show Hoechst staining of HepG2 cells treated with all tested berberine derivatives (5e, 6e and 7e) to be able to declare that berberine derivatives induce apoptosis based on these data

Reply comment:

We will analyze the Hoechst 33258 staining analysis of phototoxicity and cytotoxicity of compounds (5e and 6e) in the future.

Lane 281 – explain the reason you incubated the HepG2 cells with tested berberine derivatives for 2 hrs only when testing their effects on cell cycle distribution, versus 4 hrs in the case of phototoxicity testing, check lane 226

Reply comment:

We found that the contents of compounds 6e and 7e in the test samples were the highest in cells for 2 h. in cytometry analysis. On the other hand, in the phototoxicity test, the results were not significantly different after 2 or 4 h of incubation.

Lane 302, Fig. 7A – explain the link between cytotoxicity and photocytotoxicity of berberine derivative 5e, as it is more potent at the tested concentrations even without irradiation, but does not induce apoptosis. The cytotoxicity and photocytotoxicity is quite similar, isn´t it?  What could be a reason?

Reply comment:

We suggest that compound 5e contains phototoxic effects and cytotoxicity in HepG2 cells. Therefore, we conclude that compound 5e has both ROS-related pathways through irradiation and mitochondrial-related pathways of apoptosis by binding to mitochondrial membrane and cause the loss of mitochondrial membrane potential.

Lane 316 – this is not true for berberine derivative 5e. Based on the results of Fig. 8, there is no increase in the level of ROS in HepG2 cells treated with berberine derivative 5e.

Reply comment:

We conclude that compound 5e has both ROS-related pathways through irradiation and mitochondrial-related pathways of apoptosis by binding to mitochondrial membrane and cause the loss of mitochondrial membrane potential.

Lane 336 – explain a reason for decreasing concentrations of berberine derivatives when measuring their effect on mitochondrial membrane potential

Reply comment:

We believe that compound 5e has phototoxic effects and dark cytotoxicity.

Lane 338 – should be “mitochondrial potential” not “damaged the DNA”

Reply comment:

We corrected it in the revised manuscript.

Lane 338 to 344 – this paragraph has to be discussed within the context of apoptosis inducing activities of previously published berberine derivatives

Reply comment:

We corrected it in the revised manuscript by amending it: “These observations suggested that berberine derivatives 6e and 7e induced mitochondrial-dependent apoptosis in HepG2 by irradiation.”.

Lane 345 – results on cytochrome c release from mitochondria are not presented in the manuscript. Include the data, or rewrite the sentence.

Reply comment:

We corrected it in the revised manuscript by amending it: “So does the finding of the loss of mitochondrial potential induced by berberine (1), compounds5e, 6e and 7e at a concentration of 0.25 μM.”.

Concerning the synthesis of berberine derivatives, authors have to be more specific and specify exact time for each reaction steps, including the reflux (e.g. lane 387).

Reply comment:

We corrected it in the revised manuscript by amending it: “…the reaction mixture was refluxed for 4–8 h.”.

Lane 724 – there are no V-FITC/PI data presented in this manuscript

Reply comment:

We corrected it in the revised manuscript by amending it: “which were further confirmed by the Sub-G1 rate of cell cycle analysis.”.

Reviewer 2 Report

In this article, authors are reporting the synthesis of berberine derivatives and evaluated their cytotoxic and photocytotoxic effects. Authors claimed that the berberine derivatives with a dodecyl group at 9-/13-position could be great candidates for the anti-liver cancer medicines developments. The study is reported with sufficient data and experiments. However, article has missing important explanations, in context of the mechanism. Authors must explain possible mechanism of action that how these derivatives become better and what could be potential target of action where they are rather more effective than the parent compound, for example, the increase in the anti-cancerous property is because of increase in permeability then the sufficient assays are missing or their enhanced activity against signaling or topoisomerases etc.

Is that possible that berberine derivatives would be absorbed better in intestines or may not? “Thus, the remarkable anticancer activity of the berberine derivatives with 9-/13-position substitution of the dodecyl moiety can be attributed to their enhanced bioavailability and cell membrane permeability through the increased lipophilicity.” Membrane cell line permeability assay could provide more information to this statement. Or a reference where it has been shown that berberine derivatives have been shown enhanced bioavailability and cell membrane permeability through the increased lipophilicity, “increases with the alkyl chain length, or lipophilic characteristics” should be supported by references. Figure 2 could be with more clarity; Font size is very small to read. Legends provide no information about the experiment and data, it is written very short. Figure 7A is not clear to see, may need to present bigger size image. The synthesis of the derivatives should be shown with data besides methodology. Why is control UV irradiation is showing low level of cells damage? Missing supplemental data in submission.

Author Response

Reviewer 2-comments:

In this article, authors are reporting the synthesis of berberine derivatives and evaluated their cytotoxic and photocytotoxic effects. Authors claimed that the berberine derivatives with a dodecyl group at 9-/13-position could be great candidates for the anti-liver cancer medicines developments. The study is reported with sufficient data and experiments. However, article has missing important explanations, in context of the mechanism. Authors must explain possible mechanism of action that how these derivatives become better and what could be potential target of action where they are rather more effective than the parent compound, for example, the increase in the anti-cancerous property is because of increase in permeability then the sufficient assays are missing or their enhanced activity against signaling or topoisomerases etc.

Reply comment:

We believe that berberine derivatives 5e, 6e and 7e contains phototoxic effects in addition to general cytotoxicity. Therefore, we conclude that compound 5e has both ROS-related pathways and mitochondrial-related pathways of apoptosis. We will perform related protein analysis by Western blotting assay for berberine derivatives 5e, 6e and 7e, including Bcl-2, Bax, P53 and PARP etc., to clarify the effect of phototoxicity mechanism in the future. In addition, we currently carry out the inhibition of the degradation of supercoiled plasmid DNA (pBR322) by topoisomerase I, and whether the circular DNA can be broken down according to the ROS production induced by light irradiation of the compound, in order to understand the mechanism of photocytotoxic-related production of ROS by the berberine derivatives 5e, 6e and 7e.

Is that possible that berberine derivatives would be absorbed better in intestines or may not? “Thus, the remarkable anticancer activity of the berberine derivatives with 9-/13-position substitution of the dodecyl moiety can be attributed to their enhanced bioavailability and cell membrane permeability through the increased lipophilicity.” Membrane cell line permeability assay could provide more information to this statement. Or a reference where it has been shown that berberine derivatives have been shown enhanced bioavailability and cell membrane permeability through the increased lipophilicity, “increases with the alkyl chain length, or lipophilic characteristics” should be supported by references.

Reply comment:

Thanks very much for your reminding. Hu et al., (2015) reported that a lipophilicity substitute at 9/13-position of berberine contained the carbon chains of moderate length (twelve carbons) displayed improved lipophilicity and the strongest inhibitory effects on glioblastoma cells. Thus, berberine derivatives with moderate carbon chain length showed the most significant enhancement in lipophilicity, implying more preferable trans-membrane permeability than berberine [33].

Fu, S.; Xie,Y.; Tuo, J.; Wang, Y.; Zhu, W.; Wu, S.; Yan, G.; Hu, H. Discovery of mitochondria-targeting berberine derivatives as the inhibitors of proliferation, invasion and migration against rat C6 and human U87 glioma cells. Med. Chem. Commun. 2015, 6, 164–173.

Figure 2 could be with more clarity; Font size is very small to read. Legends provide no information about the experiment and data, it is written very short.

Reply comment:

We have resized Figure 2 and added graphical illustrations in the revised manuscript.

Figure 7A is not clear to see, may need to present bigger size image.

Reply comment:

We have resized Figure 7A and increased graphics resolution in the revised manuscript.

The synthesis of the derivatives should be shown with data besides methodology.

Reply comment:

We have illustrated the synthesis method in Scheme 1 and in Materials and Methods.

Why is control UV irradiation is showing low level of cells damage?

Reply comment:

Although a 420 nm light source is used, there are still 5% wavelengths below 400 nm, so the cells are still slightly damaged, but we believe that the results are still credible.

Missing supplemental data in submission. 

Reply comment:

We have confirmed the “Supplementary Information” summit to Molecules.

Round 2

Reviewer 1 Report

Despite authors fulfilled most of my recommendations, the manuscript still contains some discrepancies that were not corrected.

Major recommendation:

The manuscript requires the extensive editing of English language. There is still a lot of grammar errors and missing words.

Minor recommendations:

Lane 33 - 36: These two sentences in Abstract have to be corrected to make clear conclusion of the study. The phrases as “these results over and again confirmed…” and “…the test data led us to conclude” must be reformulated to make them lucid, for instance: these results confirmed that… and …based on our data we conclude that… are promising candidates for development of new anticancer agents.

Lane 300: The authors did not explain within the manuscript the reason for measuring the apoptotic effect of berberine derivatives for 2 hrs only versus 4 hrs in the case of phototoxicity testing. They replied that contents of compounds 6e and 7e in the test samples were the highest in cells for 2 h in cytometry analysis (I agree with that), and in the phototoxicity the results were not significantly different after 2 or 4 h of incubation. But these findings have to be incorporated and discussed appropriately also in the manuscript.

Lane 317-320: Fig. 7 presents results of single dose treatment. Please, include the data on concentration dependent effect of tested derivatives on sub-G1 fraction as supplemental data, or specify as “data not shown”.

Lane 339: The authors declare that “As shown in Fig. 8A, berberine (1) and compounds 5e, 6e or 7e, all stimulated generation of intracellular singlet oxygen radicals in HepG2 cells…”. Again, this is not true for berberine derivative 5e! Please, correct the sentence.

Lane 359: The authors need to be more specific and explain the reason for testing 0.25 microM berberine derivatives when measuring the effect of derivative on mitochondrial membrane potential (MMP), compare to the analysis of ROS generation and apoptosis measurement (sub-G0 fraction) where 0.5 microM concentrations were tested. Could they provide results on effect of 0.5 microM berberine derivatives on MMP?

Reviewer 2 Report

Authors responded all the queries and article would be a right fit for the publication with these improvements.